# FEATURE INCAY FOR REPRESENTATION REGULARIZATION

**Yuhui Yuan**
Microsoft Research
yuyua@microsoft.com

**Kuiyuan Yang**
DeepMotion
kuiyuanyang@deepmotion.ai

**Jianyuan Guo & Chao Zhang**
Peking University
jyguo@pku.edu.cn
chzhang@cis.pku.edu.cn

**Jingdong Wang**
Microsoft Research
jingdw@microsoft.com

## ABSTRACT

Softmax-based loss is widely used in deep learning for multi-class classification, where each class is represented by a weight vector and each sample is represented as a feature vector. Inspired by that weight decay is a common practice to regularize the weight vectors, we investigate how to regularize the feature vectors since representation is also tunable in deep learning.

One main observation is that elongating the feature norm of both correctly-classified and mis-classified feature vectors improves learning: (1) increasing the feature norm of correctly-classified examples enlarges the probability margin among different classes and ensures better generalization. (2) increasing the feature norm of mis-classified examples can up-weight the contribution from hard examples. Accordingly, we propose feature incay to regularize feature vectors by encouraging larger feature norm. Extensive empirical results on MNIST, CIFAR10, CIFAR100 and LFW demonstrate the effectiveness of feature incay.

## 1 INTRODUCTION

Deep Neural Networks (DNNs) with softmax-based loss have achieved state-of-the-art performance on numerous multi-class classification related tasks. In DNNs, both representations and classifiers are learned within a unified network concurrently, where the final representation for a sample is the feature vector $\mathbf{f}$ outputted from the penultimate layer, while the last layer outputs scores $z_i = \mathbf{w}_i \cdot \mathbf{f}$ for each category $i$, where $\mathbf{w}_i$ is the weight vector for category $i$. Before defining the loss, the scores for each category are normalized into probability via softmax function, i.e., $p_i = \frac{e^{z_i}}{\sum_j e^{z_j}}$.

A well-trained DNN should output significant larger probability for the correct label than other labels, which requires the score for the correct label is significantly larger than other labels. Since $z_i = \mathbf{w}_i \cdot \mathbf{f} = \|\mathbf{w}_i\|\|\mathbf{f}\|\cos(\theta)$, where $\theta$ is the angle between $\mathbf{w}_i$ and $\mathbf{f}$, the goal of significant larger score for the correct label than other labels can be achieved by tuning $\|\mathbf{w}_i\|$, $\|\mathbf{f}\|$ and $\theta$. While increasing the weight norm $\|\mathbf{w}_i\|$ is constrained by weight decay for regularization, thus $\|\mathbf{f}\|$ and $\theta$ become the two main factors for optimization. Although softmax loss can tune both of them, there is still much room to improve either or both factors.

Recently, Liu et al. (2016), Wang et al. (2017) and Liu et al. (2017a) propose different approaches to further optimize the factor of angular $\theta$, and all of them have achieved obviously better performance. To further emphasize the factor of angular, Ranjan et al. (2017) and Liu et al. (2017c) propose to use normalized feature vectors for softmax loss where the factor of feature norm is totally ignored. In this work, we make the effort to optimize the feature norm. Firstly, we analyze the connections between the feature norm and the classification accuracy within softmax loss, they are highly correlated as illustrated in Figure 1. Features with larger norm tend to be correctly classified with higher probability.

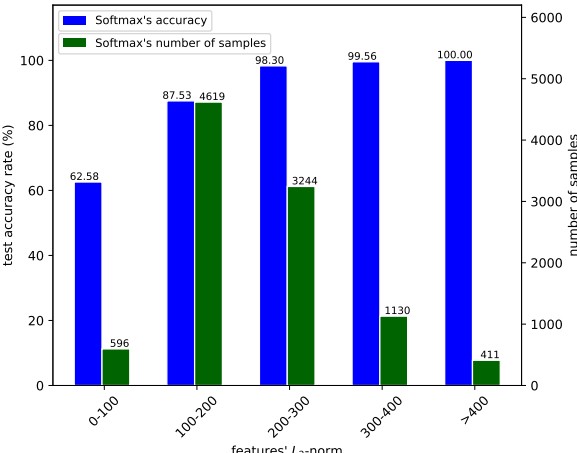

Figure 1: Test accuracy and number of samples vs Features' $L_2$-norm on CIFAR10 test set, and the model is trained with softmax loss. The test accuracy increases monotonically with the feature norm. e.g., the test accuracy reaches 100% for samples with feature norm exceeding 400.

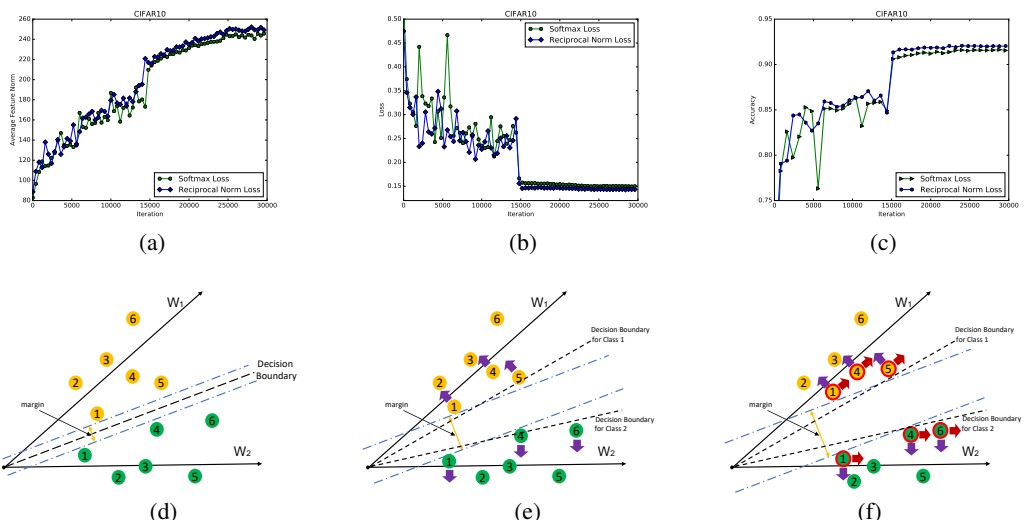

Figure 2: Comparison of Softmax loss with Softmax loss + Feature incay(i.e., Reciprocal Norm Loss, we will define it in Section 3) on test set of CIFAR10. (a) Average $L_2$-norm of feature vectors vs Iterations, (b) Softmax loss vs Iterations, (c) Top-1 accuracy vs iterations. Figure (d), (e) and (f) illustrate different approaches using binary classification as an example, where yellow points are samples of class1 and green points are samples of class2. The black dashed line represents the decision boundary between the two classes. The two blue dashed lines represent the hyperplanes that pass the points with minimal distances to the decision boundary. The numbers 1-6 represent the order of $L_2$-norm of the points within each class. $W_1$ and $W_2$ represent the weight vectors. (d) Feature embedding of Softmax loss. (e) Feature embedding of Large-margin Softmax loss. The purple arrows represent the additional angle constraints compared with Softmax loss. (f) Large-margin Softmax loss + Feature incay. The red arrows correspond to the constraints from feature incay. The margin between class1 and class2 increases from left to right.

Here we propose to optimize the feature norm by augmenting the softmax loss with feature incay. In contrast to weight decay that shrinks weight vectors to be of small norm, feature incay tends to stretch out the feature vectors. From the computational perspective, larger feature norm results in larger score differences among categories, which can better separate the categories. From the perspective of pattern detection, larger feature norm encourages model to learn and detect more prominent patterns.

Figure 2(a), 2(b) and 2(c) show the results of comparison experiments by adding feature incay to softmax loss, where feature incay achieves larger feature norm, smaller loss value and higher accuracy on test set. The geometric interpretation of feature incay is illustrated in Figure 2(d), 2(e) and 2(f), we can achieve the larger inter-class separability by explicitly optimizing both the $\|\mathbf{f}\|$ and $\theta$ compared with the other methods. Besides, the proposed feature incay (implemented as Reciprocal Norm Loss) is designed to increase the feature norm adaptively according to the original feature norm, which can also help reduce the intra-class variances as discussed in section 3.4.

Our main contributions are summarized as follows:

- The benefits of increasing the feature norm for both correctly-classified examples and misclassified examples are analyzed theoretically.
- We verify the effectiveness of feature incay empirically on four widely used classification datasets(i.e., MNIST, CIFAR10, CIFAR100 and LFW) using various network architectures. With feature incay, we achieve consistent performance improvements on all of them.
- We also conduct experiments to study the weights' distribution w/ feature incay and w/o feature incay and investigate where the improvements come from.

## 2 RELATED WORK

**Large-margin Softmax Loss.** Liu et al. (2016) proposed to improve softmax loss by incorporating an adjustable margin $m$ multiplying the angle between a feature vector and the corresponding weight vector. Compared with the softmax loss, it pays more attention to the angular decision margin between classes as illustrated in Figure 2(e). Large-margin softmax loss appends stronger constraint to the angular, while feature incay considers constraint to feature norm. As illustrated in Figure 2(f), feature incay is orthogonal to large-margin softmax loss.

**Center Loss.** Wen et al. (2016) presented the center loss to learn centers for deep features of each class and penalize the distances between the deep features and their corresponding class centers. Combining softmax loss with center loss actually uses two sets of classifiers, where representation is learned based on both the inner product to weight vector and the Euclidean distance to class center. The added center loss helps minimize the intra-class distances also by influencing the feature norm, namely, small feature norm will be increased and large feature norm will be decreased during the process of pushing feature vectors to class centers. Different from center loss, feature incay also increases the large feature norm instead of penalizing feature vectors with large norm as center loss.

**Weight/Feature Normalization.** Inspired by the fact that feature normalization before calculating the sample distances usually achieves better performance for retrieval tasks, Ranjan et al. (2017) proposed to use normalized feature vectors in softmax loss during training, thus the feature norm has no effect on softmax loss and angle is the main factor to be optimized. Congenerous cosine loss(Liu et al. (2017b)), NormFace(Wang et al. (2017)), and cosine normalization(Chunjie et al. (2017)) take a step further to normalize the weight vectors which replace inner product with cosine similarity within softmax loss, and only optimize the factor of angle. Recently, Wang et al. (2018b), Wang et al. (2018a) and Deng et al. (2018) all proposed to add the angle margin after normalizing both the weight vectors and feature vectors during nearly the same period. Although normalization mechanism achieves much lower intra-class angular variability by emphasizing more on the angle during training, they ignore that feature norm is another useful factor worth to optimize.

**Feature Scale.** COCO(Liu et al. (2017c)) and $L_2$-softmax(Ranjan et al. (2017)) are the most relevant to our work, and especially COCO is a concurrent work. Here we compare our method with COCO theoretically, *Relations*: Both COCO and feature incay increase the feature norm, which is the common reason for the performance improvement. *Differences*: COCO is optimizing the feature embedding on a hypersphere while feature incay optimizes feature embedding located between two hyperspheres with different radiuses(Property 3 in section 3.4). COCO proposes a novel con-

generous cosine loss while feature incay uses the original softmax loss. Feature incay is simpler than COCO and it can be easily plugged into almost all the related work that use softmax loss. Empirically comparision is provided in section 4.2.

# 3 OUR WORK

## 3.1 REVISITING SOFTMAX LOSS

Let $\mathbb{X} = \{(x_i, y_i)\}_{i=1}^N$ be the training set contains $N$ samples, where $x_i$ is the raw input to the DNN, $y_i \in \{1, 2, \cdots, K\}$ is the class label that supervises the output of the DNN. Denote $\mathbf{f}_i$ as the feature vector for $x_i$ learned by the DNN, $\{\mathbf{w}_j\}_{j=1}^K$ represent weight vectors for the $K$ categories. Then, softmax loss is defined as,

$$\mathcal{L}_{\text{softmax}} = -\frac{1}{N} \sum_{i=1}^N \log \left( \frac{e^{\mathbf{w}_{y_i}^T \mathbf{f}_i + \mathbf{b}_{y_i}}}{\sum_{j=1}^K e^{\mathbf{w}_j^T \mathbf{f}_i + \mathbf{b}_j}} \right) \tag{1}$$

Bias terms are ignored following the discussions in recent works Liu et al. (2016) and Wang et al. (2017). Denote the angle between $\mathbf{w}_j$ and $\mathbf{f}_i$ as $\theta_{\mathbf{w}_j, \mathbf{f}_i}$, the inner product between $\mathbf{w}_j$ and $\mathbf{f}_i$ can be rewritten as

$$\mathbf{w}_j^T \mathbf{f}_i = \|\mathbf{w}_j\| \|\mathbf{f}_i\| \cos(\theta_{\mathbf{w}_j, \mathbf{f}_i}) \tag{2}$$

By combining the above two equations, we get

$$\mathcal{L}_{\text{softmax}} = -\frac{1}{N} \sum_{i=1}^N \log \left( \frac{e^{\|\mathbf{w}_{y_i}\| \|\mathbf{f}_i\| \cos(\theta_{\mathbf{w}_{y_i}, \mathbf{f}_i})}}{\sum_{j=1}^K e^{\|\mathbf{w}_j\| \|\mathbf{f}_i\| \cos(\theta_{\mathbf{w}_j, \mathbf{f}_i})}} \right) \tag{3}$$

## 3.2 FEATURE NORM MATTERS

Here we will illustrate how feature norm influences the softmax loss from two aspects: (1) increasing the feature norm of correctly-classified examples enlarges the probability margin among different categories. (2) increasing the feature norm of mis-classified examples can up-weight the gradients of weight vectors, especially the weight vectors of the correct category and mis-classified category.

The first property is similar to the proposition proved by Wang et al. (2017), which states that softmax loss always encourages features of the correctly-classified examples to have larger magnitudes.

**Property 1** [Feature Norm Matters for Correctly-classified Examples] *Suppose weight vectors and directions of the feature vectors are fixed, increasing of the feature norm of correctly-classified examples can enlarge the probability margin among different categories.*

*Proof.* Assuming $\mathbf{f}_i$ is correctly-classified into category $y_i$, here we define the probability margin between category $y_i$ and any other category $j (j \neq y_i)$ as:

$$\mathcal{M}_{y_i, j}(\mathbf{f}_i) = P_{y_i}^i - P_j^i = \frac{e^{\mathbf{w}_{y_i}^T \mathbf{f}_i} - e^{\mathbf{w}_j^T \mathbf{f}_i}}{\sum_{k=1}^K e^{\mathbf{w}_k^T \mathbf{f}_i}}$$

$$= \frac{1 - e^{(\mathbf{w}_j^T - \mathbf{w}_{y_i}^T)\mathbf{f}_i}}{\sum_{k=1}^K e^{(\mathbf{w}_k^T - \mathbf{w}_{y_i}^T)\mathbf{f}_i}} \tag{4}$$

Recall that when $\mathbf{f}_i$ is correctly classified, we have $\mathbf{w}_{y_i}^T \mathbf{f}_i > \mathbf{w}_j^T \mathbf{f}_i$ for $j \neq y_i$. Then, for any $t > 0$, we have,

$$e^{(\mathbf{w}_j^T - \mathbf{w}_{y_i}^T)((1+t)\mathbf{f}_i)} < e^{(\mathbf{w}_j^T - \mathbf{w}_{y_i}^T)\mathbf{f}_i} \tag{5}$$

$$1 - e^{(\mathbf{w}_j^T - \mathbf{w}_{y_i}^T)(1+t)\mathbf{f}_i} > 1 - e^{(\mathbf{w}_j^T - \mathbf{w}_{y_i}^T)\mathbf{f}_i}$$

$$\sum_{k=1}^K e^{(\mathbf{w}_k^T - \mathbf{w}_{y_i}^T)(1+t)\mathbf{f}_i} < \sum_{k=1}^K e^{(\mathbf{w}_k^T - \mathbf{w}_{y_i}^T)\mathbf{f}_i} \tag{6}$$

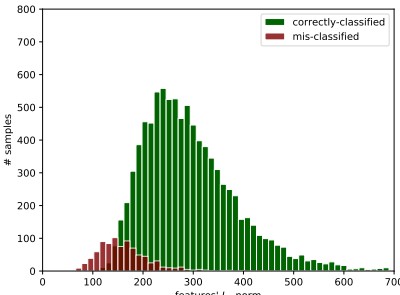

Figure 3: Distribution of features' norm over correctly-classified / mis-classified examples.

According to the above inequations, we have

$$\mathcal{M}_{y_i,j}((1+t)\mathbf{f}_i) > \mathcal{M}_{y_i,j}(\mathbf{f}_i) \tag{7}$$

$\square$

In summary, the probability margin increases by increasing the feature norm and larger probability margin ensures better generalization.

**Property 2** [Feature Norm Matters for Mis-classified Examples] *Assuming the feature vector $\mathbf{f}_i$ is mis-classified into category $k(k \neq y_i)$, increasing the feature norm of $\mathbf{f}_i$ can up-weight the gradients of the weight vector $\mathbf{w}_k$.*

***Proof.*** According to definition of softmax loss in Eq.(3), the gradient with respect to weight vector $\mathbf{w}_j$ $(j = 1, \cdots, K)$ is:

$$\frac{\partial \mathcal{L}_{\text{softmax}}}{\partial \mathbf{w}_j} = \frac{1}{N} \sum_{i=1}^{N} (P_j^i - h(i))\mathbf{f}_i \tag{8}$$

where $P_j^i = \frac{e^{\mathbf{w}_j^T \mathbf{f}_i}}{\sum_{k=1}^{K} e^{\mathbf{w}_k^T \mathbf{f}_i}}$, and $h(i)$ is an indicator function, $h(i) = 1$ if $\mathbf{y}_i = j$ otherwise $h(i) = 0$. Specifically, we analyze how the value of $\frac{\partial \mathcal{L}_{\text{softmax}}}{\partial \mathbf{w}_k}$ changes, which contains two factors: $(1)(P_k^i - h(i))$ increases with larger $\|\mathbf{f}_i\|$ when $\mathbf{f}_i$ is mis-classified as category $k$. By increasing $\|\mathbf{f}_i\|$ we will always have larger $P_k^i$ and $h(i) = 0$ when computing $\mathbf{w}_k$. So the absolute value of $(P_k^i - h(i))$ is increased. (2) $\|\mathbf{f}_i\|$ is increased. So increasing the feature norm $\|\mathbf{f}_i\|$ can up-weight the gradients for $\mathbf{w}_k$. Especially for hard examples $\mathbf{f}_i$ with $P_k^i$ close to 1 and $P_{y_i}^i$ close to 0, $(P_{y_i}^i - h(i))$ is closer to -1 by increasing $\|\mathbf{f}_i\|$, thus increasing $\|\mathbf{f}_i\|$ can also increase the gradients for weight vector $\mathbf{w}_{y_i}$. $\square$

We investigate the distribution of feature norm over correctly-classified and mis-classified examples on CIFAR10 in Figure3, and the gradients contributed by the mis-classified features is suppressed due to most of the mis-classified examples are of small feature norm. Thus it is necessary to increase the feature norm of mis-classified examples. By optimizing the feature norm of mis-classified vectors, the errors can be fixed as illustrated in the fifth column of Table 6. (e.g., 336 examples from the test set of CIFAR10 become correctly-classified by increasing their average feature norm from 169.3 to 195.9)

Besides, the proposed feature incay increases the originally small feature norm faster than the originally large feature norm. Thus the feature norm of the mis-classified examples will increase faster than the feature norm of the correctly-classified examples, which can further up-weight the contribution of the hard examples.

### 3.3 Reciprocal Norm Loss

The superiorities of increasing feature norm have been investigated in the previous subsection. Here we explore several methods that increase the feature norm end-to-end by penalizing an additional term, such as $-\|\mathbf{f}\|^2$, $-log(\|\mathbf{f}\|^2)$ and $\frac{1}{\|\mathbf{f}\|^2}$. The comparison analysis is provided in supplementary. Specifically, we choose $\frac{1}{\|\mathbf{f}\|^2}$ and propose the Reciprocal Norm Loss, where the definition of

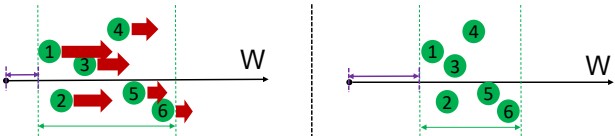

Figure 4: The original data distribution is on the left of the black dashed line and the data distribution updated according to the Reciprocal Norm Loss is on the right. The numbers 1-6 represent that the points are of increasing feature norm. The black point represents the original point. The lengths of the green bidirectional arrows represent the maximal distance in the direction of the weight vectors within all the points of one class. The purple bidirectional arrow means the minimal distance to origin, which is equal to the minimal feature norm. The red arrows represent the gradients update along the directions of the weight vectors computed with the Reciprocal Norm Loss, while the lengths represent the magnitude of the gradients.

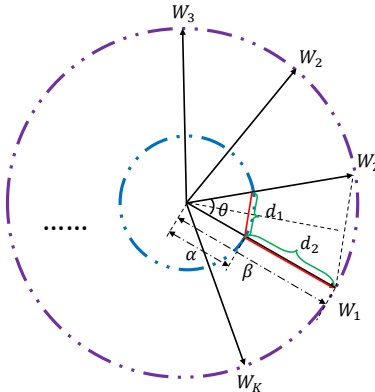

Figure 5: Illustration in 2-dimensional space. $\alpha$ and $\beta$ are the radiuses of the two circles, which represent the lower bound and upper bound of the feature norm.

Reciprocal Norm Loss is,

$$\mathcal{L} = \underbrace{-\frac{1}{N}\sum_{i=1}^{N}\log\left(\frac{e^{\mathbf{w}_{y_i}^T\mathbf{f}_i}}{\sum_{j=1}^{K}e^{\mathbf{w}_j^T\mathbf{f}_i}}\right)}_{\text{softmax loss}} + \underbrace{\mu\sum_{k=1}^{K}\|\mathbf{w}_k\|^2}_{\text{weight decay}} + \underbrace{\lambda\frac{1}{N}\sum_{i=1}^{N}\frac{1}{\|\mathbf{f}_i\|^2+\epsilon}}_{\text{feature incay}} \qquad (9)$$

where $\epsilon$ is a small positive value to prevent dividing by value close to zero, $\lambda$ is a hyper-parameter used to control the influence of feature incay. The whole loss function consists of three parts: softmax loss, weight decay and feature incay, of which the later two items prefer small weight norm and large feature norm. Feature incay can be considered during the whole training procedure or after some training iterations with softmax loss. Large feature norm brings large inter-class separability under the constraint of small weight norm, otherwise large feature norm can be trivially achieved by increasing the weight norm.

The simple reciprocal form of feature norm has an important property that moves feature vectors with small norm faster along weight vectors than feature vectors with large norm and results intra-class compactness, which is illustrated in Figure 4. Specifically, denote feature incay of $\mathbf{f}_i$ as $\mathcal{F}(\mathbf{f}_i) = \frac{1}{\|\mathbf{f}_i\|^2+\epsilon}$, the gradient is $\frac{\partial\mathcal{F}}{\partial\mathbf{f}_i} = -\frac{2\mathbf{f}_i}{(\|\mathbf{f}_i\|^2+\epsilon)^2}$. For any two feature vectors $\mathbf{f}_p$ and $\mathbf{f}_q$ satisfying $\|\mathbf{f}_q\| > \|\mathbf{f}_p\|$, we always have $\|\frac{\partial\mathcal{F}}{\partial\mathbf{f}_p}\| > \|\frac{\partial\mathcal{F}}{\partial\mathbf{f}_q}\|$. Thus vectors with small norm increase fast along their original directions while feature vectors with large norm increase slowly along their original directions.

Table 1: Error Rates (%) on MNIST/CIFAR10/CIFAR100.

| Method | MNIST | CIFAR10 | CIFAR100 |
|---|---|---|---|
| CNN(Jarrett et al. (2009)) | 0.53 | N/A | N/A |
| DropConnect(Wan et al. (2013)) | 0.57 | 9.41 | N/A |
| FitNet(Romero et al. (2014)) | 0.51 | N/A | 35.04 |
| NiN(Lin et al. (2013)) | 0.47 | 10.47 | 35.68 |
| Maxout(Goodfellow et al. (2013)) | 0.45 | 11.68 | 38.57 |
| DSN(Lee et al. (2015)) | 0.39 | 9.69 | 34.57 |
| R-CNN(Liang & Hu (2015)) | 0.31 | 8.69 | 31.75 |
| GenPool(Lee et al. (2016)) | 0.31 | 7.62 | 32.37 |
| Hinge Loss(Liu et al. (2016)) | 0.47 | 9.91 | 33.10 |
| Softmax(Liu et al. (2016)) | 0.40 | 9.05 | 32.74 |
| L-Softmax(Liu et al. (2016)) | 0.31 | 7.58 | 29.53 |
| Softmax | 0.35 | 8.59 | 32.36 |
| RN + Softmax | 0.31 | 7.84 | 31.76 |
| L-Softmax | **0.25** | 7.56 | 29.95 |
| RN + L-Softmax | 0.29 | **7.22** | **29.18** |

### 3.4 GEOMETRIC INTERPRETATION OF RECIPROCAL NORM LOSS

As stated in the previous subsection, Reciprocal Norm Loss can increase feature norm adaptively to decrease the intra-class variance. Here we prove that there exists an upper bound for the ratio between the feature norm's upper bound and the feature norm's lower bound.

**Property 3** [Feature Norm Bound] *Given (a) the angle between any feature vector $\mathbf{f}_i$ and its corresponding weight vector $\mathbf{w}_{y_i}$ is zero, (b) the angles between any two neighbor weight vectors of different classes are $\theta$, we have (1) the minimal inter-class distance is $2\alpha \sin(\frac{\theta}{2})$, where $\alpha$ is lower bound of feature norm, (2) to ensure the maximal intra-class distance is smaller than the minimal inter-class distance, the upper bound of feature norm is $3\alpha$, especially when $K < 2D(K$ is the number of classes and $D$ is the features' dimension), the upper bound in the range of $[(1 + \sqrt{2})\alpha, 3\alpha]$.*

***Proof.*** Figure 5 shows the 2-dimensional case satisfying conditions (a) and (b), where black arrows named with $W_i$ represent weight vectors for each class. As we have assumed that all feature vectors are lying on the directions of their corresponding $W_i$, blue circle and purple circle denote the lower bound and upper bound of feature norm respectively. Thus the maximal intra-class distance is $d_2 = \beta - \alpha$ and the minimal inter-class distance is $d_1 = 2\alpha \sin(\frac{\theta}{2})$. To ensure minimal inter-class distance is larger than intra-class distance, i.e., $d_1 = 2\alpha \sin(\frac{\theta}{2}) > d_2 = \beta - \alpha$, which requires $\beta < 2\alpha \sin(\frac{\theta}{2}) + \alpha \leqslant 3\alpha$. Thus $3\alpha$ is the general upper bound for the feature norm.

Especially when $K < 2D$, according to the **Lemma**(refer to supplementary), we can ensure that $\theta \geq 90°$. Besides, the angle between any two vectors is smaller than $180°$. Then $\frac{\theta}{2} \in [45°, 90°]$ and $\sin(\frac{\theta}{2})$ is a monotonously increasing function within the range $[45°, 90°]$. Based on $\sin(\frac{\theta}{2}) \in [\frac{\sqrt{2}}{2}, 1]$, the upper bound of $L_2$-norm of feature vectors is in the range $[(1 + \sqrt{2})\alpha, 3\alpha]$. $\square$

According to the **Property 3**, the ratio of the upper bound $\beta$ and the lower bound $\alpha$ is bounded($\frac{\beta}{\alpha} \leq 3$). We can estimate the upper bound of the feature norm based on the original features' $L_2$-norm. In our experiments, we choose the average $L_2$-norm as the lower bound to avoid the influence of outliers. For example, if the average $L_2$-norm on CIFAR10 is 200, we will choose $483 \approx (1 + \sqrt{2}) \times 200$ as the threshold to control the feature incay. The feature incay for features with feature norm exceeding 483 will be set as 0.

## 4 EXPERIMENTS

In this section, we verify the effectiveness of feature incay through empirical experiments.

### 4.1 EXPERIMENTAL SETTINGS

We evaluate feature incay on four datasets, i.e., MNIST, CIFAR10, CIFAR100 and LFW. MNIST consists of 60,000 training images and 10,000 test images from 10 handwritten digits, both CIFAR10

Table 2: Face verification accuracy (%) on LFW.

| Method | Data | Network | mAcc |
|---|---|---|---|
| FaceNet(Schroff et al. (2015)) | 200M | N/A | 99.65 |
| DeepID2(Sun et al. (2015)) | 300K | N/A | 99.47 |
| CenterFace(Wen et al. (2016)) | 700K | N/A | 99.28 |
| L-Softmax(Liu et al. (2016)) | CASIA-WebFace | SphereNet-64 | 99.10 |
| A-Softmax(Liu et al. (2017a)) | CASIA-WebFace | SphereNet-20 | 99.26 |
| A-Softmax(Liu et al. (2017a)) | CASIA-WebFace | SphereNet-64 | 99.42 |
| COCO(Liu et al. (2017c)) | MS-1M | ResNet-101 | **99.78** |
| COCO | CASIA-WebFace | SphereNet-20 | 98.90 |
| RN + COCO | CASIA-WebFace | SphereNet-20 | **99.02** |
| L-Softmax | CASIA-WebFace | SphereNet-20 | 99.03 |
| RN + L-Softmax | CASIA-WebFace | SphereNet-20 | **99.18** |
| A-Softmax | CASIA-WebFace | SphereNet-64 | 99.42 |
| RN + A-Softmax | CASIA-WebFace | SphereNet-64 | **99.47** |

Table 3: Accuracy(%) Comparison of different $\lambda$.

| Method | $\lambda = 1$ | $\lambda = 0.1$ | $\lambda = 0.01$ | $\lambda = 0$ |
|---|---|---|---|---|
| RN + Softmax | 91.68 | **92.16** | 91.96 | 91.41 |
| RN + L-Softmax | 92.40 | **92.78** | 92.65 | 92.44 |

and CIFAR100 contain 50,000 training images and 10,000 test images from 10 object categories and 100 object categories respectively. LFW(Huang et al. (2007)) dataset contains 13,233 face images from 5749 different identities, 6000 face pairs are used as test set following the standard protocol. Images are subtracted by mean image Images and randomly flipped horizontally for data augmentation. The specific network architectures are detailed in supplementary. We adopt Caffe framework(Jia et al. (2014)) for training and testing. The weight $\mu$ for weight decay is 0.0005 in all experiments. We choose different weight $\lambda$ in different experiments, i.e., 1.0, 0.1 or 0.01. The momentum is 0.9, and the learning rate starts from 0.1 and is divided by a factor of 10 three times when the training error stops decreasing.

## 4.2 COMPARISON EXPERIMENTS

Feature incay is added to Sofmax, L-Softmax and A-Softmax to compare with state-of-the-art approaches, which is represented with RN(Reciprocal Norm loss) plus the baseline method. e.g., RN + Softmax means combining the feature incay with Softmax loss. The reproduced results by Softmax, L-Softmax and A-Softmax following Liu et al. (2016; 2017a) are the same as or slightly better than the referred numbers in general. Table 1 reports the error rates of compared approaches and our method on MNIST, CIFAR10 and CIFAR100. It can be concluded that feature incay can consistently improve over Softmax and L-Softmax on CIFAR10/CIFAR100. For example, RN decreases the error rate of Softmax from 8.59% to 7.84% while L-Softmax achieves 7.56% on CIFAR10, which demonstrates both of them are better than Softmax. By combining RN and L-Sofmax, we achieve better result 7.22%, which means that they are reciprocal. However, RN + L-Softmax is slightly worse than L-Softmax on MNIST, our hypothesis is that the performance on MNIST is already saturate and difficult to improve further.

To further verify our method's effectiveness on more challenging datasets, we test RN + COCO, RN + L-Softmax and RN + A-Softmax on LFW and achieve competitive performance with SphereNet-20 or SphereNet-64. The results are illustrated in Table 2. The reproduced results with L-Softmax and A-Softmax are comparable while the reproduced COCO result is not as good due to both the training dataset and network structure are set different. With feature incay, RN + L-Softmax improves the L-Softmax from 99.03% to 99.18%, RN + A-Softmax improves the A-Softmax from 99.42% to 99.47% and RN + COCO improves the COCO from 98.90% to 99.02%. Thus feature incay can even promote both A-Softmax and COCO with normalized features by elongating the features before normalization.

Table 4: Accuracy(%) Comparison by Scaling the Feature.

| Scale | 1 | 2 | 4 | 6 | 8 | > 10 |
|---|---|---|---|---|---|---|
| Softmax | **91.41** | 90.90 | 90.18 | 90.91 | 90.97 | NAN |

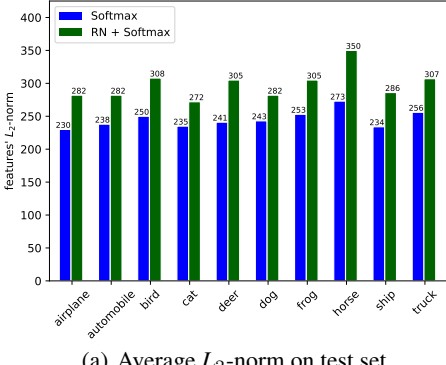

(a) Average $L_2$-norm on test set.

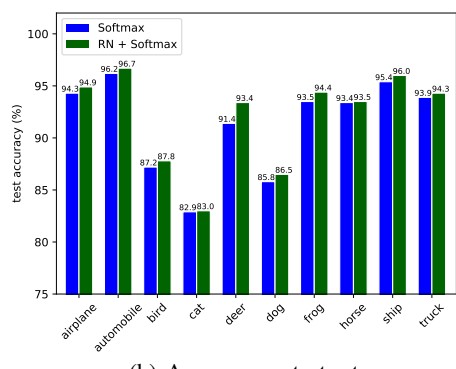

(b) Accuracy on test set.

Figure 6: Histograms of Average $L_2$-norm and Accuracy on CIFAR10, (a) the feature norm is increased over all the classes. e.g., the feature norm increases from 230 to 282 for class airplane. (b) the test accuracy is boosted over all the classes. e.g., the test accuracy increases from 95.4 to 96.0 for class ship.

Table 5: Accuracy(%) Comparison of different $\mu$ on CIFAR10. $\mu = 0.0005$ is the best choice among all of them, thus we choose this setting in all the other experiments.

| Method | $\mu = 0.00001$ | $\mu = 0.00005$ | $\mu = 0.0005$ | $\mu = 0.005$ |
|---|---|---|---|---|
| Accuracy | 89.21 | 89.35 | **91.41** | 91.16 |
| Average Feature Norm | 185.8 | 202.2 | **246.2** | 231.3 |

### 4.3 Effects of $\lambda$

Here we conduct experiments on CIFAR10 to investigate the influence of hyper-parameter $\lambda$. Results are illustrated in Table 3. Both RN + Softmax and RN + L-Softmax achieve consistent improvement for all different $\lambda$ except on RN + L-Softmax when $\lambda = 1$, which is caused by that the loss item of L-Softmax can be smaller than the loss item of feature incay. To balance the L-Softmax and feature incay for training, $\lambda$ should be set to a relatively small weight.

### 4.4 Simply Scale the Feature

To verify whether it is possible to improve the performance by simply rescaling the features before computing softmax loss, we conduct extensive experiments on CIFAR10 and present the related results in Table 4. Simply scaling the features fails to improve the performance, where too large value can cause the network fails to converge. For example, when we scale the features more than 10 times, the softmax loss will explode during training(NAN represents the softmax loss is exploded during training).

### 4.5 Effects of Weight Decay

Here we also investigate the influence of the weigth decay on the classification accuracy and feature norm, where we conduct experiments considering only Softmax loss for fairness. The results are reported in Table 5, where we find that it fails to improve neither accuracy nor feature norm by

Table 6: Average $L_2$-norm and the corresponding number of examples on test set of CIFAR10, e.g., 253.2 / 9141 represents 9141 examples are correctly classified and their average $L_2$-norm is 253.2. Error-fixed represents the examples mis-classified by Softmax but correctly-classified by RN + Softmax. Error-added represents the examples correctly-classified by Softmax but mis-classified by RN + Softmax.

| Method | Accuracy | correctly-classified | Mis-classified | Error-fixed | Error-added |
|---|---|---|---|---|---|
| Softmax | 91.41 | 253.2 / 9141 | 167.5 / 859 | 169.3 / 336 | 161.6 / 261 |
| RN + Softmax | 92.16 | 308.3 / 9216 | 187.2 / 784 | 195.9 / 336 | 187.3 / 261 |

simply increasing the weight decay or decreasing the weight decay. Thus simply changing the weight decay leads to either underfitting or overfitting.

### 4.6 Effects on Weights' Distribution

To avoid overfitting, we consider weight decay in all experiments. However, one main concern is the side effect of feature incay may increase the magnitude of the shallow layers. We plot the weights' distributions of initial state, difference choices of weight decay within Softmax and RN + Softmax in Figure 7(refer to supplementary for details of the network architectures settings). It is interesting to find that the weights of the first convolution layer(Conv0) becomes even sparser with feature incay. Overall, the influence of feature incay is limited due to the constraint from weight decay. So the feature incay enlarges the feature norm without harming the magnitude of weights from previous layers and will not lead to overfitting. Besides, we can also observe that different weight decay has big impact on the final weight distribution, such as larger weigth decay(e.g., $\mu = 0.005$) results in more weights are constrained to zero, which may lead to underfitting according to their final classification performances. More details are illustrated in Figure 7.

### 4.7 Result Analysis

By analyzing the features' $L_2$-norm and classification accuracy on CIFAR10 for data of each class, we investigate where the concrete improvements come from. Table 6 reports the related details and we find 336 examples that are mis-classified by Softmax but correctly classified by RN + Softmax. However, 261 mis-classified examples are further introduced by RN + Softmax, which limits the final performance improvement.

We also plot the histograms of average $L_2$-norm and accuracy for Softmax and Softmax + RN. The details are illustrated in Figure 6. With feature incay, the feature norm is enlarged and the accuracy is on all ten classes.

## 5 Conclusions

In this paper, we propose the feature incay implemented as Reciprocal Norm Loss to increase the feature norm. Based on the theoretical analysis of the feature norm, the Reciprocal Norm Loss enlarges the probability margin among different categories and up-weight the contribution from hard examples by managing the feature norm of both the correctly-classified and mis-classified feature vectors. Extensive experiments on MNIST, CIFAR10, CIFAR100 and LFW verify the effectiveness of our method.

## 6 Supplementary

Table 7: The CNN architectures used for MNIST/CIFAR10/CIFAR100. The count of the Conv1.x, Conv2.x and Conv3.x closely follows the settings in Liu et al. (2016). All the pooling layers are with window size $2 \times 2$ and stride of 2.

| Layer | MNIST | CIFAR10 | CIFAR100 |
|---|---|---|---|
| Conv0.x | $[3 \times 3, 64] \times 1$ | $[3 \times 3, 64] \times 1$ | $[3 \times 3, 128] \times 1$ |
| Conv1.x | $[3 \times 3, 64] \times 3$ | $[3 \times 3, 64] \times 4$ | $[3 \times 3, 128] \times 4$ |
| Conv2.x | $[3 \times 3, 64] \times 3$ | $[3 \times 3, 128] \times 4$ | $[3 \times 3, 256] \times 4$ |
| Conv3.x | $[3 \times 3, 64] \times 3$ | $[3 \times 3, 256] \times 4$ | $[3 \times 3, 512] \times 4$ |
| Fully Connected | 256 | 512 | 512 |

### 6.1 Lemma

Here is the **Lemma** proposed by Ranjan et al. (2017) and is used in the proof of **Property 3** and **Property 4**.

**Lemma** *When the number of classes $K$ is smaller than twice the feature dimension $D$, we can distribute the classes on a hypersphere of dimension $D$ such that any two class weight vectors are at least $90°$ apart.*

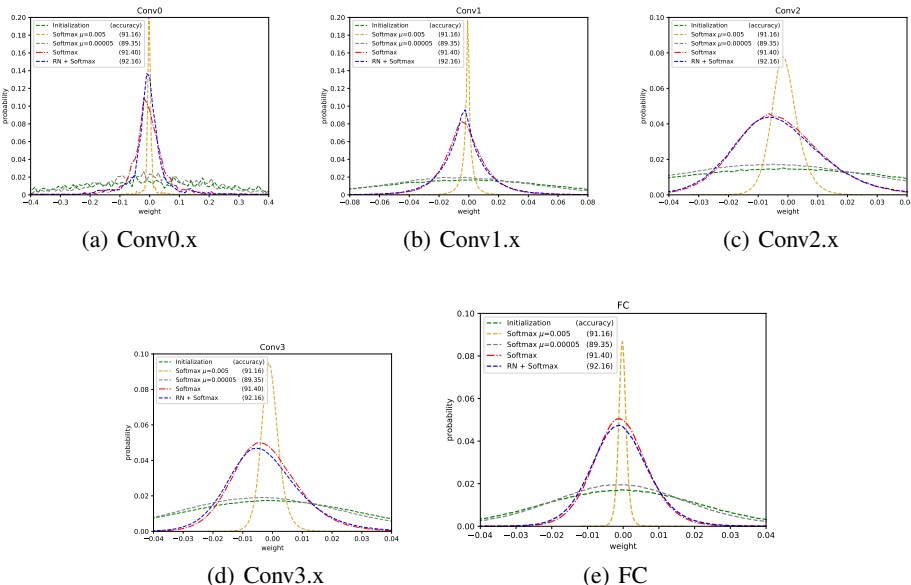

Figure 7: Histograms for Weights' distribution of different layers from model trained on CIFAR10. Here we consider five methods: (1) Weights' distribution after Initialization. (2) Weights' distribution after trained with Softmax loss where the weight decay chooses $\mu = 0.005$(classification accuracy is 91.16%). (3) Weights' distribution after trained with Softmax loss where the weight decay chooses $\mu = 0.00005$(classification accuracy is 89.35%). (4) Weights' distribution after trained with Softmax loss where the weight decay chooses $\mu = 0.0005$(classification accuracy is 91.40%). (5) Weights' distribution after trained with RN + Softmax(classification accuracy is 92.16%). The magnitude of the weight parameters is only slightly influenced by the feature incay. Besides, Softmax $\mu = 0.005$ represents larger weigth decay while Softmax $\mu = 0.00005$ represents smaller weight decay compared with the standard settings.(e.g., $\mu = 0.0005$). The weights parameters are very sparse within Softmax $\mu = 0.005$ while very dense within Softmax $\mu = 0.00005$, which induce either underfitting or overfitting.

## 6.2 FEATURE NORM WITHIN SOFTMAX LOSS

Here we mainly want to present you that the feature norm of correctly classifed examples fails to be increased within softmax loss, while feature incay can increase the feature norm consistently.

**Property 4** [Feature Norm in Softmax Loss]*For any feature vector* $\mathbf{f}_i$, *if* $P^i_{y_i} \to 1$ *and* $P^i_j \to 0 (\forall j \neq y_i)$, *then* $\frac{\partial \mathcal{L}_{softmax}}{\partial \mathbf{f}_i} \to 0$.

***Proof.*** According to definition of softmax loss in Eq.(3), the gradient of feature vector $\mathbf{f}_i$ is:

$$\frac{\partial \mathcal{L}_{\text{softmax}}}{\partial \mathbf{f}_i} = \frac{1}{N}(-\mathbf{w}_{y_i} + \sum_{j=1}^{K} P^i_j \mathbf{w}_j) \tag{10}$$

where the $P^i_j = \frac{e^{\mathbf{w}_j^T \mathbf{f}_i}}{\sum_{k=1}^{K} e^{\mathbf{w}_k^T \mathbf{f}_i}}$. When $P^i_{y_i} \to 1$ and $P^i_j \to 0(\forall j \neq y_i)$, $\frac{\partial \mathcal{L}_{\text{softmax}}}{\partial \mathbf{f}_i} \to 0$. That is after a training sample is confidently classified correctly, it will have no contribution to its own representation learning. For example, suppose $\|\mathbf{w}_j\| = 1$, $\theta_{\mathbf{w}_{y_i}, \mathbf{f}_i} = 0$ and $\theta_{\mathbf{w}_j, \mathbf{f}_i} \geq 90°(\forall j \neq y_i)$ according the **Lemma**, then $\|\mathbf{w}_{y_i}\|\|\mathbf{f}_i\| \cos(\theta_{\mathbf{w}_{y_i}, \mathbf{f}_i}) = \|\mathbf{f}_i\|$, $\|\mathbf{w}_j\|\|\mathbf{f}_i\| \cos(\theta_{\mathbf{w}_j, \mathbf{f}_i}) < 0, \forall j \neq y_i$. Putting together, we have $P^i_{y_i} \geq \frac{e^{\|\mathbf{f}_i\|}}{e^{\|\mathbf{f}_i\|} + K - 1}$, for a modest number of categories say $K = 10$, $P^i_{y_i} > 0.999$ when $\|\mathbf{f}_i\| = 10$. □

## 6.3 FUNCTIONAL FORMAT OF FEATURE INCAY

The choice of the functional format of feature incay is important. Here we mainly analyze three different choices.

Table 8: Results of comparison experiments for three feature incay on CIFAR10, where LN represents the linear form, LogN represents the log form and RN represents the reciprocal form.

| Method | Softmax | Softmax + LN | Softmax + LogN | Softmax + RN |
|---|---|---|---|---|
| Accuracy | 91.41 | 91.86 | 91.83 | **92.16** |

- **Linear.** $\mathcal{F}(\|\mathbf{f}\|^2) = -\|\mathbf{f}\|^2$ $\quad \frac{\partial \mathcal{F}}{\partial \mathbf{f}} = -2\mathbf{f}$

- **Log.** $\mathcal{F}(\|\mathbf{f}\|^2) = -log(\|\mathbf{f}\|^2)$ $\quad \frac{\partial \mathcal{F}}{\partial \mathbf{f}} = -\frac{2\mathbf{f}}{\|\mathbf{f}\|^2}$

- **Reciprocal.** $\mathcal{F}(\|\mathbf{f}\|^2) = \frac{1}{\|\mathbf{f}\|^2}$ $\quad \frac{\partial \mathcal{F}}{\partial \mathbf{f}} = -\frac{2\mathbf{f}}{\|\mathbf{f}\|^2\|\mathbf{f}\|^2}$

Here we mainly analyze the differences of the above three functions by investigating the relationship between the gradients and the original feature norm. Assuming that we have two features $\mathbf{f}_1$ and $\mathbf{f}_2$ satisfying $\|\mathbf{f}_2\| > \|\mathbf{f}_1\|$. For the linear function case, we have $\| \frac{\partial \mathcal{F}}{\partial \mathbf{f}_2} \| > \| \frac{\partial \mathcal{F}}{\partial \mathbf{f}_1} \|$. Then, the intra-class variance will be increased as the distance between $\mathbf{f}_1$ and $\mathbf{f}_2$ increases with each gradient update. Besides, the gradients of the linear function have the same magnitude with the feature itself, such large gradients update can lead to explosion during the training. For the log function case, we have $\frac{\partial \mathcal{F}}{\partial \mathbf{f}} = \frac{2\mathbf{f}}{\|\mathbf{f}\|^2} = \frac{2\mathbf{u}}{\|\mathbf{f}\|}$, where $\mathbf{u}$ is the unit vector with feature norm equals 1. Thus $\| \frac{\partial \mathcal{F}}{\partial \mathbf{f}_1} \| > \| \frac{\partial \mathcal{F}}{\partial \mathbf{f}_2} \|$. The gradients within reciprocal function is also that $\| \frac{\partial \mathcal{F}}{\partial \mathbf{f}_1} \| > \| \frac{\partial \mathcal{F}}{\partial \mathbf{f}_2} \|$ always holds once $\|\mathbf{f}_2\| > \|\mathbf{f}_1\|$. Although both log function and reciprocal function increase the features with small feature norm faster than the features with larger feature norm, we find the performance of reciprocal function is better. In summary, reciprocal function can increase the overall feature norm and increase the intra-class similarity simultaneously, where the intra-class similarity along the direction of the weight vectors can be decreased with the log function. We choose the reciprocal function in all of our experiments.

Table 8 reports the classification accuracies adopting the three different considered feature incay. The superiority of Softmax + RN over Softmax + LN and Softmax + LogN is well illustrated, where Softmax + RN can achieve better intra-class similarity according to the above analysis.

### 6.4 CNN ARCHITECTURES SETTINGS

For LFW, we adopt 20-layer/64-layer SphereNet following the same settings in Liu et al. (2017a). We modify the network settings for MNIST/CIFAR10/CIFAR100 based on the previous work(Liu et al. (2016)) and list them in Table 7.

### 6.5 EXPERIMENTAL ANALYSIS

Figure 8(a), 8(b) and 8(c) show the accuracy, average feature norm and softmax loss during training by using $\lambda = 1, 0.1, 0.01$ respectively on CIFAR10. Feature incay achieves better or comparable accuracy compared with softmax loss under a wide range of $\lambda$, and results in larger feature norm on both training and test set. All methods achieve close to zero softmax loss on training set, while feature incay ensures lower softmax loss on test set. Figure 8(d) shows the accuracy vs iteration number on CIFAR100, which is similar to the results on CIFAR10.

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

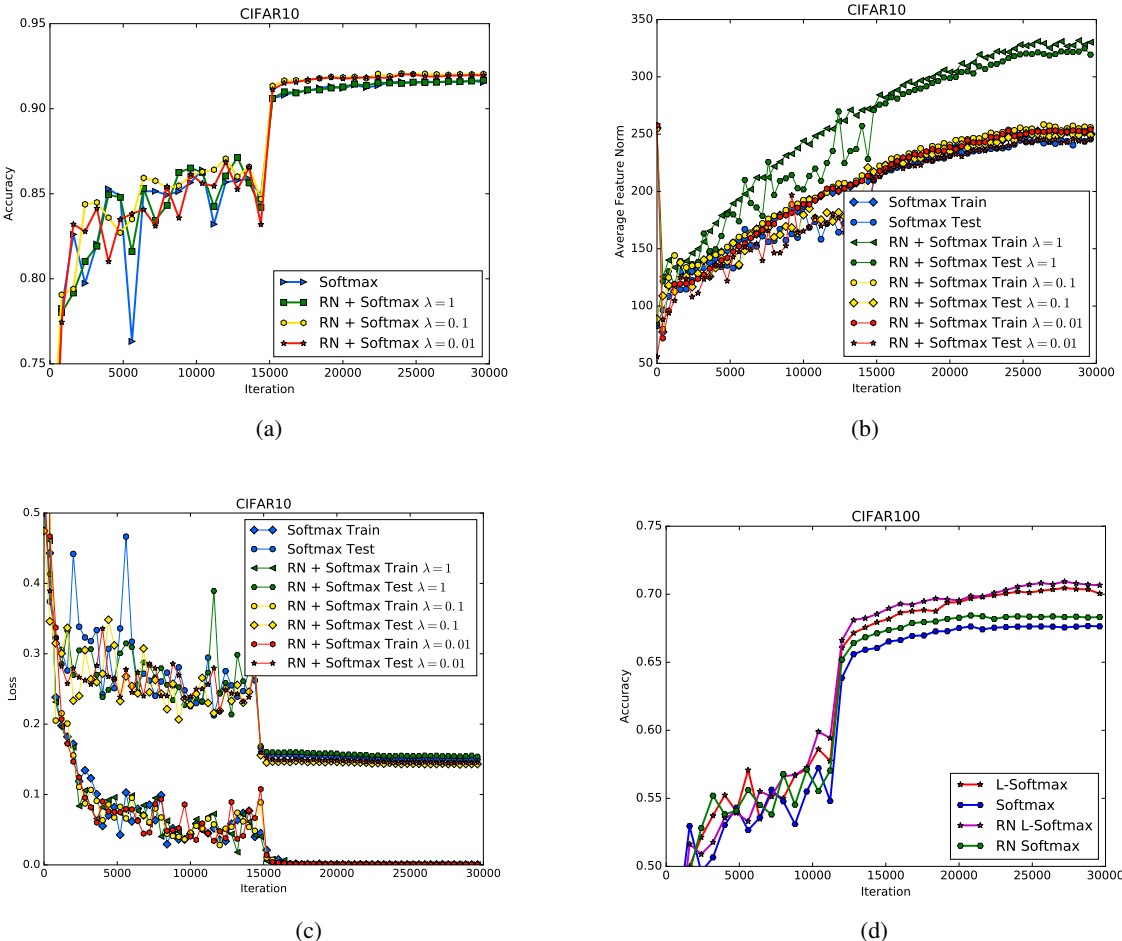

Figure 8: (a) Accuracy versus iterations with different choices of $\lambda$ value on the test set of CIFAR10. The RN Softmax achieves 92.04% when $\lambda = 0.1$ (b) The training/testing sets' $L_2$ norm vs iterations with different choices of the $\lambda$ value on CIFAR10. The RN Softmax with different $\lambda$ all achieve larger $L_2$-norm. (c) The training/testing sets' loss vs iterations with different choices of the $\lambda$ value on CIFAR10. The RN Softmax achieves notable smaller loss value 0.1432 than the Softmax with 0.1498. The training loss is very small for all the methods, but RN + Softmax has significantly smaller testing loss.(It is best viewed by zooming the figure.) (d) Accuracy vs iterations with Softmax/RN Softmax/L-Softmax/RN L-Softmax on CIFAR100. Both RN Softmax and RN + L-Softmax achieve better performance compared with baseline, where the best method is RN + L-Softmax.

Kevin Jarrett, Koray Kavukcuoglu, Yann LeCun, et al. What is the best multi-stage architecture for object recognition? In *Computer Vision, 2009 IEEE 12th International Conference on*, pp. 2146–2153. IEEE, 2009.

Yangqing Jia, Evan Shelhamer, Jeff Donahue, Sergey Karayev, Jonathan Long, Ross Girshick, Sergio Guadarrama, and Trevor Darrell. Caffe: Convolutional architecture for fast feature embedding. In *Proceedings of the 22nd ACM international conference on Multimedia*, pp. 675–678. ACM, 2014.

Chen-Yu Lee, Saining Xie, Patrick Gallagher, Zhengyou Zhang, and Zhuowen Tu. Deeply-supervised nets. In *Artificial Intelligence and Statistics*, pp. 562–570, 2015.

Chen-Yu Lee, Patrick W Gallagher, and Zhuowen Tu. Generalizing pooling functions in convolutional neural networks: Mixed, gated, and tree. In *International conference on artificial intelligence and statistics*, 2016.

Ming Liang and Xiaolin Hu. Recurrent convolutional neural network for object recognition. In *Proceedings of the IEEE Conference on Computer Vision and Pattern Recognition*, pp. 3367–3375, 2015.

Min Lin, Qiang Chen, and Shuicheng Yan. Network in network. *arXiv preprint arXiv:1312.4400*, 2013.

Weiyang Liu, Yandong Wen, Zhiding Yu, and Meng Yang. Large-margin softmax loss for convolutional neural networks. In *Proceedings of The 33rd International Conference on Machine Learning*, pp. 507–516, 2016.

Weiyang Liu, Yandong Wen, Zhiding Yu, Ming Li, Bhiksha Raj, and Le Song. Sphereface: Deep hypersphere embedding for face recognition. *arXiv preprint arXiv:1704.08063*, 2017a.

Yu Liu, Hongyang Li, and Xiaogang Wang. Learning deep features via congenerous cosine loss for person recognition. *arXiv preprint: 1702.06890*, 2017b.

Yu Liu, Hongyang Li, and Xiaogang Wang. Rethinking feature discrimination and polymerization for large-scale recognition. 2017c.

Rajeev Ranjan, Carlos D Castillo, and Rama Chellappa. L2-constrained softmax loss for discriminative face verification. *arXiv preprint arXiv:1703.09507*, 2017.

Adriana Romero, Nicolas Ballas, Samira Ebrahimi Kahou, Antoine Chassang, Carlo Gatta, and Yoshua Bengio. Fitnets: Hints for thin deep nets. *arXiv preprint arXiv:1412.6550*, 2014.

Florian Schroff, Dmitry Kalenichenko, and James Philbin. Facenet: A unified embedding for face recognition and clustering. In *Proceedings of the IEEE Conference on Computer Vision and Pattern Recognition*, pp. 815–823, 2015.

Yi Sun, Xiaogang Wang, and Xiaoou Tang. Deeply learned face representations are sparse, selective, and robust. In *Proceedings of the IEEE Conference on Computer Vision and Pattern Recognition*, pp. 2892–2900, 2015.

Li Wan, Matthew Zeiler, Sixin Zhang, Yann L Cun, and Rob Fergus. Regularization of neural networks using dropconnect. In *Proceedings of the 30th International Conference on Machine Learning (ICML-13)*, pp. 1058–1066, 2013.

Feng Wang, Xiang Xiang, Jian Cheng, and Alan L Yuille. Normface: $l_2$ hypersphere embedding for face verification. *arXiv preprint arXiv:1704.06369*, 2017.

Feng Wang, Weiyang Liu, Haijun Liu, and Jian Cheng. Additive margin softmax for face verification. *arXiv preprint arXiv:1801.05599*, 2018a.

Hao Wang, Yitong Wang, Zheng Zhou, Xing Ji, Zhifeng Li, Dihong Gong, Jingchao Zhou, and Wei Liu. Cosface: Large margin cosine loss for deep face recognition. *arXiv preprint arXiv:1801.09414*, 2018b.

Yandong Wen, Kaipeng Zhang, Zhifeng Li, and Yu Qiao. A discriminative feature learning approach for deep face recognition. In *European Conference on Computer Vision*, pp. 499–515. Springer, 2016.

