# OpenReview forum: "Feature Incay for Representation Regularization"
_ICLR.cc/2018/Conference — Invite to Workshop Track_

### Official Review · AnonReviewer2 · 2017-11-27
**This paper investigates how to finetune feature norms of correctly-classified and mis-classified feature vectors to improve learning process. Based on the analysis, they proposed feature incay to encourage larger feature norm. Experimental results on four datasets demonstrate the effectiveness of the proposed method.**

**Rating:** 6
**Confidence:** 2

**Review:**

Pros:
1. It provided theoretic analysis why larger feature norm is preferred in feature representation learning.

2. A new regularization method (feature incay) is proposed.

Cons:
It seems there is not much comparison between this proposed method and the concurrent work "COCO(Liu et al. (2017c))".

---

> ### Author Response · Authors · 2017-12-06
> **Comparison with  "COCO(Liu et al. (2017c))".**
>
> Thanks a lot for your positive and constructive comments!
>
> We provide the response to "It seems there is not much comparison between this proposed method and the concurrent work 'COCO(Liu et al. (2017c))'."
>
> (1) Both "COCO" and "Feature Incay" increase the L2-norm of feature representations, which is the common reason for performance improvement.
>
> (2) There are several clear differences.
>      a. "COCO" normalizes and rescales all features to have the same L2-norm while"Feature Incay" adds a new regularizer that prefers features with larger L2-norm. "COCO" uses the optimal scale value that is fixed during training while "Feature Incay" increases the feature norm without constraining the scale value.
>     b. “COCO” optimizes feature embedding spreading on a hypersphere while “Feature Incay” optimizes feature embedding located between two hyperspheres with different radiuses. (see Property 3)
>     c. "COCO" proposes a novel congenerous cosine loss while "Feature Incay" uses the original softmax loss:  "Feature Incay" is simpler than "COCO"  and it can be easily plugged into almost all the related works that use softmax loss.
>
> (3) We compare the "COCO" with "RN + COCO" on CASIA-WebFace with SphereNet-20 and find that "Feature Incay" can help improve the performance of "COCO". e.g., "RN + COCO" improves "COCO" from 98.90% to 99.02%.

---

### Official Review · AnonReviewer3 · 2017-11-27
**The idea is interesting, but motivation requires more justification. Results are good, but not very impressive**

**Rating:** 6
**Confidence:** 4

**Review:**

The manuscript proposes to increase the norm of the last hidden layer to promote better classification accuracy. However, the motivation is a bit less convincing. Here are a few motivations that are mentioned.
(1) Increasing the feature norm of correctly classified examples helps cross entropy, which is of course correct. However, it only decreases the training loss. How do we know it will not lead to overfitting?
(2) Increasing the feature norm of mis-classified examples will make gradient larger for self-correction. And the manuscript proves it in property 2. However, the proof seems not complete. In Eq (7), increasing the feature norm would also affect the value of the term in parenthesis. As an example, if a negative example is already mis-classified as a positive, and its current probability is very close to 1, then further increasing feature norm would make the probability even closer to 1, leading to saturation and smaller gradient.
(3) Figure 1 shows that examples with larger feature norm tend to be predicted well. However, it is not very convincing since it is only a correlation rather than causality. Let's use simple linear softmax regression as a sanity check, where features to softmax are real features rather than hidden units. Increasing the feature norm seems to be against the best practice of feature normalization in which each feature after normalization is of variance 1.

The manuscript states that the feature norm won't be infinitely increased since there is an upper bound. However, the proof of property 3 seems to only apply to the certain cases where K<2D. In addition, alpha is in the formula of upper bound, but what is the upper bound of alpha?

The manuscript does comprehensive experiments to test the proposed method. The results are good, since the proposed method outperforms other baselines in most datasets. But the results are not impressively strong.

Minor issues:
(1) For proof of property 3, it seems that alpha and beta are used before defined. Are they the radius of two circles?

---

> ### Author Response · Authors · 2017-12-11
> **Justification about our motivation and other issues**
>
> Thanks a lot for your insightful comments.
>
> -1- Increasing the feature norm of correctly classified examples helps cross entropy, which is of course correct. However, it only decreases the training loss. How do we know it will not lead to overfitting?
>
> Good question. In our experiments, we don't find the feature incay will lead to overfitting. e.g., by considering feature incay, RN + Softmax decreases the training loss and improves the Softmax from 91.41% to 92.16% on the test set of CIFAR10. It remains an open problem to provide theoretical analysis about whether increasing the feature norm will lead to overfitting currently.
>
>
> -2-  Increasing the feature norm of mis-classified examples will make gradient larger for self-correction. And the manuscript proves it in property 2. However, the proof seems not complete. In Eq (7), increasing the feature norm would also affect the value of the term in parenthesis. As an example, if a negative example is already mis-classified as a positive, and its current probability is very close to 1, then further increasing feature norm would make the probability even closer to 1, leading to saturation and smaller gradient.
>
> Thanks for pointing this problem. The proof of property 2 is indeed complete.  In your described case, increasing the feature norm will not lead to smaller gradients for both the weight vectors of the ground truth category and the wrongly predict category , which instead will have larger gradients.
>
> We give the reasons below. For a mis-classified sample i with ground truth label y_i.  It is true that "When the mis-classified f_i has probability of class k(k!=y_i) close to 1, then increase the feature norm of f_i will make the probability of class k even closer to 1", but this will not cause "saturation and smaller gradient" for all w_k and w_(y_i).  According to Equation (7):
>
> (1) the gradient of w_(y_i) : when  j=y_i, h(i)=1, P_j^i is close to 0, then (P_j^i-h(i)) is close to -1, so the gradients for the weight vector of ground truth category  can be increased by increasing the norm of f_i;
> (2) the gradient of w_(k): when j=k, h(i)=0, as that P_k^i is close to 1, (P_k^i-h(i)) is close 1, the gradients for weight vector of the wrongly predict category can be increased by increasing the norm of f_i.
> (3) the gradients of other w_j:  when j!=y_i && j!=k ,  (P_j^i-h(i)) is close to 0, thus the gradients is close to zero.
>
>
>
> -3- Figure 1 shows that examples with larger feature norm tend to be predicted well. However, it is not very convincing since it is only a correlation rather than causality. Let's use simple linear softmax regression as a sanity check, where features to softmax are real features rather than hidden units. Increasing the feature norm seems to be against the best practice of feature normalization in which each feature after normalization is of variance 1.
>
> Thanks for pointing out this interesting problem.  As we observe that the feature norm and the classification accuracy is positively related,  and we investigate whether increasing the feature norm explicitly could improve the performance and find that the classification accuracy is improved with the feature incay.  It also remains an open problem to provide theoretical analysis about whether  it is correlation or causality currently.
> Increasing the feature norm is not against the best practice of feature normalization.  In fact, increasing the feature norm before normalization can also help improve the final performance, which is shown in Table 1 and stated in the last sentence of Section 4.2.("feature incay can even promote the A-softmax with normalized features by elongating the features before normalization")
>
>
>
> -4- The manuscript states that the feature norm won't be infinitely increased since there is an upper bound. However, the proof of property 3 seems to only apply to the certain cases where K<2D. In addition, alpha is in the formula of upper bound, but what is the upper bound of alpha?
>
> Thanks for pointing out this issue. Our property essentially is not limited to K<2D. We updated Property 3 for both K<2D and K>=2D case:
> "...(2) to ensure the maximal intra-class distance is smaller than the minimal inter-class distance, the upper bound of feature norm is 3*alpha, especially when K < 2D, the upper bound in a tighter range of [(1 + sqrt(2))*alpha, 3*alpha]". So 3*alpha is a general upper bound whether K<2D or K>=2D. Especially, when K<2D, we can formulate a tighter range for the upper bound.
>
> What's the upper bound of alpha is an interesting problem, but it is not our current interest. The main point of Property 3 lies in that the ratio of the upper bound beta to the lower bound alpha is bounded: beta/alpha <= 3.
>
> -5- For proof of property 3, it seems that alpha and beta are used before defined. Are they the radius of two circles?
>
> Yes, they are the radius of the two circles.

---

### Official Review · AnonReviewer1 · 2017-11-27
**Interesting analysis of feature norm, but the paper needs improvements**

**Rating:** 6
**Confidence:** 3

**Review:**

The analyses of this paper (1) increasing the feature norm of correctly-classified examples induce smaller training loss, (2) increasing the feature norm of mis-classified examples upweight the contribution from hard examples, are interesting. The reciprocal norm loss seems to be reasonable idea to improve the CNN learning based on the analyses.

However, the presentation of this paper need to be largely improved. For example, Figure 3 seems to be not relevant to Property2 and may be show the feature norm is lower when the samples is hard example. Therefore, the author used reciprocal norm loss which increases feature norm as shown in Figure 4. However, both Figures are not explained in the main text, and thus hard to understand the relation of Figure 3 and 4. The author should refer all Figures and Tables.

Other issues are:
-Large-margin Soft max in Figure 2 is not explained in the introduction section.
-In Eq.(7), P_j^I is not defined.
- In the Property3, The author wrote “ where r is lower bound of feature norm”.
 However, r is not used.
-In the experimental results, “RN” is not defined.
-In the Table3, the order of \lambda should be increasing or decreasing order.
- Table 5 is not referred in the main text.

== Updated review ==
The presentation has been improved, I have increased the rate from 5 to 6.
Following are further comments for presentation.

-	Fig.2 “ the increasing L2 norm “ seems to  “the order of L2 norm ”
-	Pp.4 the first sentence above Eq.(7) “According to definition …”  should be improved .
-	pp.5, the first sentence of the second paragraph “The feature norm can be optimized ..” is not clear.
-	It would be better put Figure 5 under Property3.
-	D should be defined in Property3.
-	pp.8 wrote “However, 259-misclassfied examples are further introduced”. However, in Table 5, it seems to be 261.
-	Section 5. is “Conclusion and future work”. However, future work is not mentioned.

---

> ### Author Response · Authors · 2017-12-11
> **Update on the presentation issues of our paper.**
>
> Thanks for your comments.
>
> -1- The presentation of this paper need to be largely improved.
> We have improved the presentation of our paper and updated the pdf files according to your advice.
>
> -2- Figure 3 seems to be not relevant to Property 2 and may be show the feature norm is lower when the samples is hard example.
> Actually, Figure 3 is relevant to Property 2. We revised the description and re-plot Figure 3 in the paper to make their relation much clearer and avoid the possible misunderstanding.
>
> We provide a short explanation below. The purpose of Figure 3 is to show that the mis-classified examples(we can also call them "hard examples") tend to be of small feature norm, which has been re-plot based on your advice. Property 2 is proposed to state that we need to increase the feature norm of mis-classified examples(tend to with small feature norm), which makes larger gradient and helps correcting the mis-classified examples.
> Especially, the fifth column in Table 5 shows that by increasing the feature norm of mis-classified examples, the "RN + Softmax" correctly classifies 336 examples that are mis-classified by "Softmax".
>
> -3- The author used reciprocal norm loss which increases feature norm as shown in Figure 4. However, both Figures are not explained in the main text, and thus hard to understand the relation of Figure 3 and 4.
> Thanks for pointing out this problem. We now added the explanations in the main text. Figure 4 is used to show the Reciprocal Norm Loss can result in more intra-class compactness by increasing the small feature norm faster than the large ones. Figure 3 is not related to Figure 4, and it is about Property 2.
>
> -4- Large-margin Soft max in Figure 2 is not explained in the introduction section.
> Thanks for pointing out this problem. We provided the explanation of Large-margin Softmax loss in the first paragraph of Section 2 (Related work). We will put it to the introduction section if it is necessary.
>
> -5- In Eq.(7), P_j^I is not defined.
> We have added the definition of P_j^i in Eq.(7) in the updated paper. In fact, we have also defined P_j^i in Property 4.
>
> -6- In the Property 3, The author wrote “ where r is lower bound of feature norm”. However, r is not used.
> Thanks for pointing out this problem, which is a typo. "r" should be replaced with "alpha".
>
> -7- In the experimental results, “RN” is not defined.
>  "RN" refers to the feature incay with form of Reciprocal Norm. We have added that "... RN(Reciprocal Norm loss) plus the baseline method. e.g., RN + Softmax means combining the feature incay with Softmax loss." in the updated paper.
>
> -8- In the Table 3, the order of \lambda should be increasing or decreasing order.
> We have resorted it in decreasing order.
>
> -9- Table 5 is not referred in the main text.
> Thanks for pointing out this problem. We have discussed the results in Table 5 in the first paragraph in Section 4.5 and added the reference to Table 5 in the updated paper.

---

### Decision · Program_Chairs · 2018-01-29
**ICLR 2018 Conference Acceptance Decision**

**Decision:**

Invite to Workshop Track

**Comment:**

 + An intriguing novel regularization method: encouraging larger norms for the feature vector input to the last softmax layer of a classifier.
 + Resonably extensive experimental validation shows that it improves test accuracy to some degree.
 - While a motivation is given, the formal analysis of what is really going on remains very superficial and limited.

Technical note: Simply scaling the softmax layer's input would not change class rankings, so any positive effect of this regularizer on classification performance is due to it changing the learning dynamic in the upper layers as well. The paper could be much stronger if it did provide an analysis regarding how the global learning dynamic is affected in all layers, by the interaction between weight decay and the last layer's feature incay.

---

> ### Author Response · Authors · 2018-02-03
> **Justification about the technical note**
>
> Thanks for your comments.
>
> "The paper could be much stronger if it did provide an analysis regarding how the global learning dynamic is affected in all layers, by the interaction between weight decay and the last layer's feature incay."
>
> We have always contained the analysis you mentioned in the supplementary meterials of our paper.
>
> Please check Section 6.6 and Figure 7.